# Fucoidan Structure and Activity in Relation to Anti-Cancer Mechanisms

**DOI:** 10.3390/md17010032

**Published:** 2019-01-07

**Authors:** Geert van Weelden, Marcin Bobiński, Karolina Okła, Willem Jan van Weelden, Andrea Romano, Johanna M. A. Pijnenborg

**Affiliations:** 1Faculty of Science, (Medical) Biology, Radboud University, 6525 XZ Nijmegen, The Netherlands; 2The First Department of Gynecologic Oncology and Gynecology, Medical University of Lublin, 20-081 Lublin, Poland; m.s.bobinski@gmail.com (M.B.); karolinaokla@gmail.com (K.O.); 3Department of Obstetrics & Gynecology, Radboud University Nijmegen, Medical Centre, 6525 GA Nijmegen, The Netherlands; willemjan.vanweelden@radboudumc.nl; 4Department of Obstetrics and Gynecology, GROW-School for Oncology and Developmental Biology Maastricht University Medical Centre, 6229 HX Maastricht, The Netherlands; a.romano@maastrichtuniversity.nl

**Keywords:** fucoidan, structure, cancer, natural product, brown algae, AKT, MAPK

## Abstract

Fucoidan is a natural derived compound found in different species of brown algae and in some animals, that has gained attention for its anticancer properties. However, the exact mechanism of action is currently unknown. Therefore, this review will address fucoidans structure, the bioavailability, and all known different pathways affected by fucoidan, in order to formulate fucoidans structure and activity in relation to its anti-cancer mechanisms. The general bioactivity of fucoidan is difficult to establish due to factors like species-related structural diversity, growth conditions, and the extraction method. The main pathways influenced by fucoidan are the PI3K/AKT, the MAPK pathway, and the caspase pathway. PTEN seems to be important in the fucoidan-mediated effect on the AKT pathway. Furthermore, the interaction with VEGF, BMP, TGF-β, and estrogen receptors are discussed. Also, fucoidan as an adjunct seems to have beneficial effects, for both the enhanced effectiveness of chemotherapy and reduced toxicity in healthy cells. In conclusion, the multipotent character of fucoidan is promising in future anti-cancer treatment. However, there is a need for more specified studies of the structure–activity relationship of fucoidan from the most promising seaweed species.

## 1. Introduction

For centuries, plants have been used in traditional medicines in the treatment and prevention of different diseases. Multiple anticancer agents in clinical use, are derived from plants such as, paclitaxel, vinblastine, and camptothecin [1,2]. The different conditions in marine environments result in a huge stock of potential bioactive compounds that are in general easily tolerated by the human body, resulting in few side-effects [3]. The advantages of natural products compared to synthetic include; lower development costs, widespread accessibility, and potential reduced side effects. Currently, it is estimated that more than 60% of the anticancer drugs are derived from plants, bacteria, and marine organisms [4].

Fucoidan is a natural derived compound found in different species of brown algae (Phaeophyceae) and in some animals, that has gained attention for its anticancer properties. Fucoidan is a sulfated polysaccharide molecule, and part of the cell wall of algae [5,6]. Algae are an exceptionally diverse group of organisms. In reference to biomass, brown algae are the dominant organism in many coastal regions and have evolved independently from each other, exhibiting many novel features [7]. Despite their phylogenetic proximity to brown algae, red and green algae do not have fucoidan in their cell walls [8]. In East Asia, brown seaweed has been utilized in the local cuisine and as a medicine for centuries [9]. Epidemiological studies have reported that the incidence of chronic diseases, such as heart disease, diabetes, and cancer is lower in China and Japan than in western countries, which may be attributed to differences in lifestyle and diet [10]. This has led to a renewed interest in the properties and contents of brown algae. In the past 50 years, more than 3000 products, derived from algae, have been discovered [11]. The polysaccharides have attracted much attention and have been deemed the most promising for their anticancer activities [12,13,14,15].

Over the course of time, various reviews have reported on fucoidans bioactivity and structure. The published reviews can be distinguished into two groups. The first is mainly based on the structure and the purification techniques [16] The second is mainly activity-based, for instance, the influence of fucoidan on inducing apoptosis and reducing migration [17]. This review will focus on the structure–activity relationship, the main cellular pathways affected by fucoidan, and the cell-specific effect of fucoidan. Subsequently, the existing data will be translated to determine the potential relevance of fucoidan in cancer treatment. Finally, challenges to clinical implementation of fucoidan-based cancer treatment are defined.

## 2. Summary of Literature

Despite the growing interest in fucoidan, there are only a few studies concerning the structure–activity relationship. In this section the main findings of the different studies concerning the structure of fucoidan (Section 2.1), the molecular weight (Section 2.2), the sulfate groups (Section 2.3), and the pharmacokinetics (Section 2.4) will be summarized. Subsequently, the cellular pathways (Section 2.5) and receptors (Section 2.6) that can interact with fucoidan are summarized.

### 2.1. Structure of Fucoidan

The structure of fucoidan depends highly on the algae species, but it always contains a backbone of sulfated fucans. In some species, the sulfated fucans backbone contains branching, consisting of different sugars, fucose, or uronic acid. Because of the branching and the heterogeneous biochemical properties, it is very difficult to study the molecule as a whole. Many studies limit their structural research on highly purified fractions, limiting our understanding on the bioactivity of the whole molecule [18]. The backbone of fucoidan does not cover the whole structure of the compound, as it is far more complex and has diverse branching. Yet, the backbone is often used as reference and to classify the molecules. Furthermore, the structure–activity relationship of the backbones is more obvious. Two types of fucoidan molecules can be distinguished. A backbone of (1 → 3)-linked α-l-fucopyranose residues (type 1, Figure 1A) or alternating (1 → 3)-linked α-l-fucopyranose and (1 → 4)-linked α-l-fucopyranose residues (type 2, Figure 1B) [19]. The structure of fucoidan from *Fucus vesiculosus* is the best-studied, being a relatively simple structure consisting mainly of fucose and sulfate branching [20] (Figure 1C). Fucoidans, extracted from *F. vesiculosus*, contains a backbone of α-(1 → 3)-linked fucose and α-(1 → 4) linked fucose residues. Sulfation occurs mainly at *O*-2 and at a lesser extent at *O*-3. Also, 2, 3-*O*-disulfate fucose residues were sometimes found [21,22]. The sulfate groups and the 2, 3-*O*-disulfate fucose, in particular, are important for the bioactivity of fucoidan molecules [16].

For a long time, it was thought that fucoidan consists of pure sulfated fucose residues (and some traces of other sugars) until the structure of fucoidan from *Macrocystis pyrifera* was found to have a heteropolymer of fucose, galactose, and trace xylose [16]. The presence of neutral sugars was also discovered in other fucoidans. The presence of these sugars resulted in increased complexity of structural analysis [16].

The reported structures of fucoidan from different species of brown algae resulted in an improved categorization of the structures. For instance, most of the fucoidans from species belonging to the Fucales order show an alternating linkage of (1 → 3)-α-l-fucose-(1 → 4)-α-l-fucose [20,23,24,25,26]. The structures of *Ascophyllum nodosum* [27] and *F. vesiculosus* resemble each other, only differing in sulfation patterns and the presence of glucuronic acid in *A. nodosum*. Most fucales species like *Fucus distichus, Pelvetia canaliculata* and *Fucus serratus* have a similar backbone but are more diverse in the branching and the presence of different sugars [24,25,28]. However, there are exceptions, for instance, fucoidans from *Himanthalia elongata* and *Bifurcaria bifurcata* do not follow this trend [29]. Thus, it seems challenging to identify the structure of fucoidan based on the order they belong to.

Also, the structure of fucoidan is dependent on the harvest season. Fucoidan from *Undaria pinnatifida* showed distinct characteristics and bioactivity when harvested in different conditions and seasons [41,42].

Moreover, the structure of fucoidan is dependent on the purification method. New purification techniques led to the discovery that the structure of fucoidan consisted of multiple fractions. Besides the major components, consisting of linked fucose-residues, also smaller fractions were noted, consisting of neutral sugars [20]. A study reported that the structure of crude fucoidan from *A. nodosum* was a predominant repeat of [→(3)-α-l-Fuc(2SO^3−^) - (1 → 4)-α-l-Fuc(2,3diSO^3−^)-(1)]n [21]. But a purified fraction from the same species consisted of primarily α-(1 → 3)-fucosyl residues with a sparse α-(1 → 4) linkage and being highly branched [33]. Different extraction techniques lead to different structures. Importantly; one species was reported to produce two distinct different fucoidan structures, namely galactofucans and uronofucoidans [32]. Therefore, the purification method is an important determinant of the structure and the related bioactivity. Additionally; some brown algae species contain multiple different fucoidan structures.

### 2.2. Molecular Weight

The molecular weight is relevant in anticancer effects, as high molecular weight fucoidan is often more effective than low molecular weight fucoidan. They are classified into: low molecular weight fucoidan (LMWF) (<10 kDa), middle molecular weight fucoidan (MMWF) (10–10,000 kDa), and high molecular weight fucoidan (HMWF) (>10,000 kDa) [43]. LMWF (*Cladosiphon navae-caledoniae)* is able to induce apoptosis in breast cancer cell lines [44]. In bladder cancer cell lines, LMWF (*Sargassum hemiphyllum*) inhibited angiogenesis through interaction with the HIF-1a/VEGF signaling pathway [45]. The proliferation of T24 bladder cancer cell lines, implanted in mice, showed to be inhibited by LMWF treatment. LMWF has been shown to enhance the therapeutic effect of chemotherapy when applied in combination and reduce the side effects [46]. The angiogenetic effect of fucoidan is molecular weight-dependent. LMWF (<15 kDa) induced angiogenesis on human umbilical vein endothelial cells [47,48], whereas HMWF (30 kDa) showed an inhibitory effect [49].

The final bioactivity of LMWF is dependent on the type of extraction method. Acidic hydrolysis is often used, but due to the loss of branching, the bioactivity is reduced [17]. Interestingly, enzymatic degradation of HMWF resulted in a more bioactive LMWF, due to retaining of the sulfate groups, that could alternatively be obtained by gamma-irradiation with even better anticancer properties [50]. LMWF might have more favorable pharmacokinetically properties, however, the degradation of crude fucoidan to LMWF often leads to loss of bioactivity.

In conclusion: the molecular weight is directly linked to fucoidans bioactivity and to the extraction method.

### 2.3. Sulfate Groups

Both the sulfate content and the position of the sulfate groups are important for the bioactivity [16,51]. Studies report that the presence of different other sugars like galactose or xylose, are only important because of the attached sulfate groups [16]. A fucoidan fraction from *Sargassum fusiforme* with a molecular weight of 12.4 kDa and 7.5% of sulfate content, was unable to inhibit the angiogenesis of HMEC-1 cells. However, a larger fraction with higher sulfate content (MW: 47.5 kDa and 20.8% sulfate), showed an inhibitory effect on the angiogenesis of HMEC-1 cells [52].

Low and high molecular weight fucoidans (*U. pinnatifida)* were chemically modified to yield more sulfate groups. The oversulfated LMWF (56.8%) was the most effective at inhibiting the growth of cancer cells. Furthermore, the authors suggested that LMWF is more suitable for oversulfation due to less steric hindrance, compared to HMWF [53]. The oversulfation of fucoidan (*F. vesiculosus)* resulted in a stronger inhibition of angiogenesis [54]. Sulfated extracts from the same species exerted higher antiproliferative effects on breast cancer cell lines [55]. It was suggested that the oversulfation resulted in an increased negative charge that was responsible for the increased bioactivity [56].

In addition to the content of sulfate groups, also the position is relevant for the bioactivity [57].

Most of the sulfate groups in *S. cichorioides, F. evanescens and Saccharina japonica* are in axial positions [58], determining the conformational flexibility of fucoidan [59].

Thus, the sulfate content and position of the sulfate groups are important determinants for fucoidans bioactivity.

### 2.4. Pharmacokinetics

There are only a few studies addressing the absorption, distribution, metabolism, and excretion (ADME) of fucoidan. Since the molecular weight of crude fucoidan is relatively high, absorption is low, as confirmed by ELISA with fucoidan-specific antibody in humans [21,60].

Fucoidan (*F. vesiculosus*, 737 kDa) absorption was studied in rats, where a maximum concentration was reached after 4 h. Most of the absorbed fucoidan accumulated in the kidney, which has repeatedly been confirmed by others [61,62,63]. Importantly, accumulation in the kidney was confirmed by others that studied the absorption of fucoidan (*Cladosiphon okamuranus)* in rats. The authors suggested that a small portion of fucoidan was absorbed via endocytosis [64]. Based on the fact that the molecular weight of absorbed fucoidan (*C. okamuranus*) was not altered it can be concluded that fucoidan is not degraded by enzymes nor that the molecular weight is altered by bacterial flora in the intestine [65,66]. However, the acidic conditions in the stomach are able to hydrolyze fucoidan in a limited fashion, explaining the small fraction of altered fucoidan found in urine [65]. In healthy volunteers, the limited absorption of orally administered fucoidan to the bloodstream was noted [65]. Currently there are two ongoing clinical trials, concerning the tolerance and biodistribution of fucoidan(-like) compounds. The safety, biodistribution, and dosimetry of a labelled fucoidan compound is tested in healthy humans volunteers (Clinicaltrials.gov, Bethesda, MD, USA, NCT03422055). In the other ongoing trial, the addition of fucoidan to chemotherapy is studied in patients with stage III-IV Non-Small Cell Lung Cancer (NSCLC) in a placebo-controlled trial to determine impact on quality of life Clinicaltrials.gov (Bethesda, MD, USA, NCT03130829). Outcomes of these clinical trials will gain insight in the toxicity and ADME of fucoidan in humans.

In animal models, even relative high repeated doses of fucoidan did not trigger a toxic response [67,68]. This is in accordance with the *in vitro* results of testing fucoidan on healthy human cells from other studies [50,69,70,71].

Due to the large molecular weight fucoidan has a poor absorption rate when administered orally. An alternative to the oral administration fucoidan can also be administered intravenously. In rabbits, the intravenous administration of low-molecular-weight fucoidan (50 mg/kg of body weight) led to a rapid absorption [72]. Otherwise, low molecular weight fucoidan (LMWF) may be developed for clinical purposes. In a comparative study on the absorption of LMWF and MMWF (*S. japonica),* it was demonstrated that LMWF had better absorption rate and bioavailability then MMWF supporting its potential [49].

Fucoidan has favorable pharmacokinetics, in reference to toxicity. However, there is not much known on the biodistribution in humans. Animal models show low bioavailability, raising interest in developing LWMF as a potential solution.

### 2.5. Cellular Mechanisms in Relation to Fucoidan Anti-Cancer Activity

In this section, all the reported pathway and receptors affected by fucoidan are summarized. Due to cross-interaction between pathways and different receptors, there will be overlap between the sections. Fucoidan is a multipotent molecule interacting with various cancer-related cellular pathways. The most important reported pathways: PI3K/AKT, the MAPK, and the caspase pathway are described in detail.

#### 2.5.1. The PI3K/AKT Pathway

The phosphatidylinositol-4,5-bisphosphate 3-kinase/protein kinase B (PI3K/AKT) signaling pathway is a central player in diverse cellular functions. This pathway is often upregulated in tumor cells and is often targeted in cancer treatment. The ability of tumor cells to resist cell death and to migrate has been connected with an upregulation of the PI3K/AKT pathway [73,74,75,76,77,78,79,80,81]. Studies have found that the PI3K/AKT pathway is involved in stimulation of the expression of different matrix metalloproteinases (MMPs) in cancer cells [82,83,84,85]. These MMPs are involved in the breakdown of extracellular matrix (ECM), and are often expressed in metastatic cancer cells [82,83,84,85,86]. In various cancer types, the interaction between fucoidan and the kinase AKT has been reported (Figure 2, Table 1).

In colon carcinoma cell lines, the protein levels of phosphorylated AKT were reduced in vitro, after fucoidan treatment, resulting in apoptosis [87]. In bladder cancer cell lines, fucoidan (*F. vesiculosus*) inhibited the phosphorylation of the PI3K/AKT pathway in vitro, resulting in apoptosis and inhibition of telomerase activity [88].

Fucoidan (*U. pinnatifida*) inhibited the phosphorylation of PI3K/AKT in prostate cancer cell lines in vitro [89]. AML cell lines showed a reduction of phosphorylated AKT in vitro, upon treatment with fucoidan [90]. The same reduction in AKT was seen in two other AML cell lines (NB4 and HL60) when treated with fucoidan in vitro, derived from *F. vesiculosus* [69]. A fucoidan extract from *F. vesiculosus* was tested on different female cancer cell lines (breast-, ovarian-, uterine-, endometrial carcinoma) in vitro. In most cancer cell lines, fucoidan treatment resulted in a decrease in phosphorylated PI3K, AKT and mTOR. The reduced activity of mTOR led to autophagy in the tested cell lines [91].

In highly metastatic lung cancer cell lines, fucoidan (*F. vesiculosus*) showed an inhibitory effect on the migration and invasion by reducing the expression of MMP-2 in vitro. Western blot revealed a decrease in phosphorylated protein levels of PI3K, AKT and mTOR [92]. When hepatocarcinoma cells in mouse were treated with purified fucoidan from *U. pinnatifida* (UPS), PI3K/AKT signaling was inhibited in vitro, resulting in a reduced metastasis [93].

In urinary bladder cancer cell lines, fucoidan (*F. vesiculosus*) inhibited MMP-9 expression without affecting AKT protein levels in vitro [88]. However, the protein levels of MMP-9 was reduced. Fucoidan-treated bladder cancer cell lines showed a decrease in MMP-9 expression but, at the same time, an increase in phosphorylated AKT protein level in vitro [94]. In the same study, the activity of NF-kB was reduced in vitro, after treatment with fucoidan. NF-kB is the upstream target of AKT and is able to stimulate MMP-9 expression [95]. This suggests that fucoidan is able to inhibit the activity of NF-kB in an AKT-independent manner.

In colorectal cancer, fucoidan induced upregulation of AKT and expression of p21WAF1 expression in vitro. It was suggested that the upregulated AKT signaling was vital for the induction of the cell cycle inhibitor p21WAF1 [96]. Similar results were seen, when bladder cancer cells were treated with fucoidan in vitro. The upregulation of AKT signaling led to the induction of p21WAF1, leading to the subsequent inhibition of the cell cycle [94]. In HCT-15 cells, fucoidan treatment led to an increase in AKT phosphorylation after 6 h, resulting in the generation of reactive oxygen species (ROS) in vitro. There are some studies reporting that an upregulated AKT signaling pathway leaves the cell more vulnerable for ROS induction [97,98]. This can be a potential fucoidan-mediated pathway.

As stated before, the regulation of the AKT pathway is complex. One of the negative regulators of AKT is the protein phosphatase and tensin homolog (PTEN), being a potential target of fucoidan. To examine this potential interaction, the PTEN expression of the affected cancer cell lines was retrieved from existing literature [99,100,101,102,103,104,105,106,107,108,109,110,111,112]. Almost all the cancer cell lines who were vulnerable for fucoidan, showed PTEN expression, except for PC-3 (bladder cancer) [101], MES-SA (uterine sarcoma) [102], and RL95-2 (endometrial cancer) [103]. These three cell lines are PTEN-deficient, but their growth is still inhibited by fucoidan. The ovarian carcinoma, Caov-3 cell line, is the only cell line unaffected by fucoidan while having wild-type PTEN expression [104]. The interaction of fucoidan with PTEN is probably just one of the different regulators of the PI3K/AKT pathway.

Some cancer cells seem to be largely insensitive for the effect of some types of fucoidan. Different uveal melanoma cancer cell lines are able to resist the anti-proliferative effect of fucoidan (*F. vesiculosus*) [113]. The underlying mechanism for their immunity remains unclear. All the cell lines contain a mutation in their G-protein coupled receptor rendering them constitutively in their active GTP state [114]. This results in the overactivation of MAPK, PKC, and PI3K/AKT pathways [115]. The inhibitory effect of fucoidan appears to be insufficient to have a significant effect on the proliferation of these cancer cells.

Thus, it becomes clear that fucoidan is able to interact with the PI3K/AKT pathway, decreasing both cell proliferation and migration, however not all cancer types are affected. The fucoidan-mediated inhibition of PI3K/AKT pathway can partially be explained by direct stimulation of PTEN. The PI3K/AKT pathway is a central player in fucoidan-mediated inhibition of various cancer cell lines.

#### 2.5.2. MAPK Signaling Pathway

The MAPK/ERK pathway is often dysregulated in cancer cells, therefore it is the subject of many studies to identify the pathway as a potential target in cancer treatment [116]. There are multiple components of MAPK/ERK pathway, that are often mutated in different types of cancer (Figure 2) [117]. Furthermore; the MAPK/ERK pathway is also involved in cell migration and invasion. The p38 MAPK and JNK pathways are also involved in tumorigenesis [118,119,120,121]. The best-studied is the MAPK/ERK pathway and fucoidan-mediated effects on this pathway are often mentioned. Some studies have reported fucoidan-induced regulation on p38 MAPK, but their relationship is less obvious. Here, we will discuss these the ERK1/2 and the p38 MAPK pathways (Table 1).

In a human lymphoma cell line, fucoidan (*F. vesiculosus*) decreased the protein level of phosphorylated ERK1/2 in vitro, seemingly crucial for the induction of apoptosis [122]. Similar to the PI3K/AKT pathway, ERK1/2 is able to prevent the cell from undergoing apoptosis. In breast cancer cell lines, fucoidan *(F. vesiculosus*) induced apoptosis, as well as reducing the protein levels of ERK1/2, survivin and Bcl-2 in vitro [123]. Fucoidan had no effect on healthy mouse fibroblasts in vitro. Fucoidan reduced, in vitro, the expression of ERK1/2 in an acute promyelocytic leukemia cell line [69]. In lung cancer cell lines, fucoidan from *F. vesiculosus* and *S. Japonica* reduced the level of active ERK1/2 in vitro. The fucoidan-induced growth inhibition can thus be partially attributed to the negative regulation of fucoidan on ERK1/2 in vitro [124]. In prostate cancer cell line, fucoidan (*U. pinnatifida*) exerted also an apoptotic effect in vitro. Western blot showed, that the fucoidan-induced apoptosis paired with the reduction in ERK1/2 and AKT. The same results were noted in vivo [125]. Fucoidan (*F. vesiculosus*) induced apoptosis in a human mucoepidermoid carcinoma cell line, by inhibiting ERK1/2 in vitro [126].

Upon fucoidan (*F. vesiculosus)* treatment, the ability to migrate and invade in lung cancer cell lines was inhibited in vitro. The inhibition was associated with a decrease in ERK1/2 and PI3K/AKT/mTOR signaling. As both pathways are linked with MPP-9 and MPP-2 expression [82,83,124], it remains unclear what their individual effect is on migration and invasiveness. Human keratinocyte cell line showed a decreased ultraviolet-B (UVB) induced expression of MMP-1 after treatment with fucoidan (*Costatia costata*) in vitro [127]. Earlier studies showed that the inhibition of ERK1/2 leads to a reduced MMP-1 expression [128].

In an attempt to uncover the cellular mechanism of fucoidan-mediated inhibition of ERK1/2, all the known mutations and receptor expression of the different types of cancer cells were assessed [95,129,130,131]. However, there was no clear mutation or absence of a particular receptor which was connected with the failure to inhibit the ERK1/2 expression. For some cancer cell lines, we found clues in the existing literature that might serve as an explanation. For instance, U937 leukemia cells have constitutive low MAPK activity and have proven to be more resistant for MAPK-targeting therapies [129]. It was not possible to extrapolate it to the other unaffected cancer cells.

Fucoidan (*F. vesiculosus)* increased the expression of p38 MAPK in colon carcinoma cell lines in vitro, resulting in the induction of apoptosis [87]. In a leukemia cell line, the activation of the p38 MAPK pathway was crucial in the induction of apoptosis mediated by fucoidan (*F. vesiculosus*) treatment [132]. In gastric cancer cell lines, treatment with fucoidan (*C. okamuranus)* decreased expression of apoptosis signal-regulating kinase 1 (ASK1) and led to a decrease in phosphorylated p38 MAPK in vitro. ASK1 is genetically upregulated in these cells and stimulates cell proliferation via the cell cycle [133]. ASK1 is downstream activated by p38 MAPK. This suggests that fucoidan exert his effect via the p38 MAPK pathway [134]. Thus, both stimulation and inhibition of the p38 MAPK pathway leads to reduced proliferation. The same fucoidan (*F. vesiculosus*) was used in both of the studies on leukemia and colon cancer cells, whereas another type of fucoidan (*Cladosiphon okamuranus*) was used on the gastric cancer cells. Thus, the differences in the effect on p38 MAPK phosphorylation is due to fucoidans structural differences and differences in cancer cell characteristics [20,23,24,25,26].

The role of p38 MAPK in fucoidan-induced cell cycle arrest is further uncovered with a study on hepatocarcinoma cell lines. Fucoidan (*F. vesiculosus*) induced cell cycle arrest in HepG2 by stimulating cell cycle inhibitors p16^INK4a^-pRb and p14^Arf^-p53 via p38 MAPK stimulation in vitro [135].

However, there are other studies reporting that fucoidan had no effect on the phosphorylation of p38 MAPK [91,94].

In conclusion, fucoidan can induce the inhibition of ERK1/2 and, to a lesser extent, p38 MAPK. This effect is dependent on cancer cell line and fucoidan species.

#### 2.5.3. The Caspase Pathway

The caspase pathway plays a central role in both the intrinsic and extrinsic induction of apoptosis. Upon activation, by cleavage, caspase will activate the apoptotic pathway. Caspases are cleaved by the apoptosome complex, consisting of cytochrome C and Apaf- [137]. Cytochrome C (Cyt c) is released from the mitochondria, regulated by multiple members of the Bcl-2 family (Figure 2) [138,139]. In an in vitro study on human mucoepidermoid carcinoma cell lines, fucoidan (*F. vesiculosus*) induced apoptosis by activation of caspase-3 [126]. Similar effect was noted when human lymphoma cell lines were treated in vitro with different concentrations of fucoidan (*F. vesiculosus*). The loss of mitochondrial membrane potential (MMP) and the subsequent release of Cyt c caused the activation of caspase 3 [122]. In mouse breast cancer, fucoidan (*F. vesiculosus*) induced the release of Cyt c in vivo and in vitro [123]. Fucoidan-induced apoptosis in human breast cancer cells by cleavage of caspase-3. The same effect was noted in vivo in mice [140]. Melanoma cells showed to be sensitive to fucoidan (*Sargassum henslowianum* and *F. vesiculosus*), inducing apoptosis by cleavage of caspase-3 in vitro [141].

As demonstrated (Figure 2), the caspase pathway is connected with both the PI3K/AKT and the MAPK pathway. By decreasing activity of these pathways, the pro-apoptotic members of the Bcl-2 family are activated [142,143,144]. Fucoidan-induced apoptosis is often associated with a decrease in PI3K/AKT and/or MAPK signaling. Furthermore, a study reported on fucoidan activating the caspase pathway independently from PI3K/AKT or MAPK [145]. Fucoidan is able to activate the caspase pathway, proving the multipotent character of fucoidan.

### 2.6. Growth-Involved Receptors

The MAPK and the PI3K/AKT pathways are the main pathways affected by fucoidan. Besides, the importance of the fucoidan-induced regulation of cellular organization, the receptor interaction also draws attention. The importance of receptor expression has been recognized in cancer research. Nowadays, there are multiple drugs that are targeting the receptors and consequently the underlying cellular mechanism [146,147,148]. Fucoidan is able to inhibit the expression of different proteins and receptors in cancer cells as shown in the following section.

#### 2.6.1. Transforming Growth Factor Beta

The cytokine transforming growth factor beta (TGF-β) plays a central role in cancer-related processes like metastasis and tissue invasion [149]. Multiple studies have reported the inhibitory effect of fucoidan on transforming growth factor receptors. In human renal cell lines, treatment with fucoidan inhibited, in vitro, the activation of TGF-β1 and decreased the receptor binding of TGF-β1. HMWF, from *F. vesiculosus* was more effective than the purified and low molecular weight form of fucoidan in this study [150,151]. Fucoidan (*S. hemiphyllum*) decreased in vitro TGF-β signaling in hepatocellular carcinoma cell lines, resulting in the inhibition of metastasis. [152]. In a triple negative breast cancer cell line, treatment with fucoidan, from *F. vesiculosus,* accelerated the ubiquitin-dependent degradation of TGF-β receptors I and II (TGF-RI and TGF-RII) in vitro [153]. In non-small cell lung cancer cell lines, fucoidan stimulates the ubiquitin-dependent degradation in vitro of TGF-RI/II via Smurf2/Smad7 [154].

In human foreskin fibroblast, the expression of TGF-RII was increased after treatment with fucoidan in vitro (*S. hemiphyllum)*. Importantly, the authors tested 4 different fractions of fucoidan with different molecular weight and sulfate content. The two fractions with the highest sulfate content were the most potent stimulators of the TGF-RII signaling [50]. Thus, it is clear that fucoidan is able to inhibit the TGF-β signaling pathway both in vitro and in vivo. Furthermore, in healthy cells fucoidan appears to have a stimulatory effect on TGF-β signaling.

#### 2.6.2. Bone Morphogenetic Protein

Bone morphogenetic proteins (BMPs) belongs to the large family of transforming growth factor-beta (TGF-β). It is a subfamily of cytokines, involved in the regulation of bone formation and frequently upregulated in cancer cells [155]. A HMWF fraction of *Nemacystus decipiens* inhibited angiogenesis via the inhibition of BMP4 in vitro in human mesenchymal endothelial cells. The expression of BMP4 was significantly reduced, and the phosphorylation of the Smad pathway was reduced [156]. However, fucoidan (*S. japonica*) stimulated in vitro osteoclast differentiation in human mesenchymal stem cells. The expression of BMP2 was increased, just as the activation of the Smad pathway was increased. Importantly, the activation of BMP2 was dependent on the phosphorylation of JNK and ERK1/2 [157]. Accordingly; fucoidan (*U. pinnatifida*) was able to stimulate osteoblast differentiation in MG-63 osteosarcoma cell line, by increasing the expression of BMP2 in vitro [158]. LMWF (*S. hemiphyllum)* increased the expression of BMP2 and other osteoblast differentiation markers in vitro and in vivo [159].

The effects of fucoidan on BMP expression seems cell-specific and, importantly, molecular weight-dependent. It is known that fucoidan interacts with BMP expression and its downstream pathway.

#### 2.6.3. Vascular Endothelial Growth Factor

Vascular endothelial growth factors (VEGFs) are often associated with cancer growth and angiogenesis. The VEGF pathway is heavily embedded in the PI3K/AKT and MAPK signaling pathways (Figure 2) [160,161,162].

Various studies report both inhibitory and stimulatory effects of fucoidan treatment on VEGF expression and its downstream effectors. Fucoidan (*U. pinnatifida* extract) inhibits in vitro the formation of micro vessels in a human umbilical vein endothelial cell line by inhibiting the expression of VEGF-A isotype [163]. In 4T1 mouse cancer cell line, fucoidan (*F. vesiculosus*) inhibited in vitro the expression of VEGF and reduces angiogenesis in vivo [123]. Fucoidan (*F. vesiculosus*) reduced the in vitro expression of VEGF in immortal retinal pigment epithelium cell lines. Importantly, it showed a synergistic effect when combined with bevacizumab, a monoclonal antibody against VEGF [164]. In mice, injected with lung cancer cells, prophylactic fucoidan (*F. vesiculosus*) treatment resulted in reduced metastasis by inhibiting VEGF expression. Also, there were no cytotoxic effects on healthy cells noted [70]. Fucoidan (*U. pinnatifida*) was shown to inhibit VEGFR3 expression in human lymphatic endothelial cells, resulting in the subsequent inhibition of downstream effectors and migration. Western blot analysis reveals a decreased AKT and NF-kB protein expression. The inhibitory effect on the lymphangiogenesis was noted both in vitro and in vivo [165]. Fucoidan was able to reduce the expression of HIF-1a under hypoxia conditions, resulting in a reduction of VEGF expression in multiple myeloma cell lines in vitro [166]. Sulfated fucoidan extract (*S. fusiforme*) inhibited migration in human microvascular endothelial cells in vitro, by competitively inhibiting the interaction of VEGF and VEGFR2. The sulfated fraction FP08S2, with a weight of 47.5 kDa, was the most effective at inhibiting the tube formation of human mesenchymal endothelial cells 1 (HMEC-1) in vitro. Importantly, the desulfated derivative, with similar molecular weight, failed to inhibit the tube formation [167].

In vivo lung cancer cells, sulfated fucoidan extract was able to inhibit the tumor growth in vitro but via the reduction of expression of VEGF or HIF-1a [168]. To mimic the in vivo situation; MG63 osteosarcoma cell line was co-cultured with human endothelial cells (OECs). Fucoidan treatment (*F. vesiculosus*) resulted in reduced expression levels of VEGF and other pro-angiogenic factors (SDF-1, Ang-2). The overall result was an insignificant reduction in bone formation, possibly due to the dosage [169].

Fucoidan (*F. vesiculosus*) had no significant influence on different uveal melanoma cell lines. The VEGF expression in OMM2.5, 92.1, and Mel270 cells was not reduced after fucoidan treatment, and only reduced at high concentrations of fucoidan in OMM2.3 and OMM1 cells [113]. Another study reported the opposite effect of low molecular weight fucoidan (LMWF) on VEGF expression in a human umbilical vein endothelial cell line. The binding of VEGF165 to different VEGF receptors was strongly increased upon fucoidan treatment [170]. Furthermore, fucoidan (*S. japonica)* stimulated in vitro osteoblast differentiation by inducing the expression of VEGF in mesenchymal stem cells [171].

Fucoidan reduces VEGF signaling in a wide variety of cancer types both in vitro and in vivo, without cytotoxic effects on healthy cells.

#### 2.6.4. Estrogen Receptor

Estrogen has proliferative effect in hormone-dependent cancers such as breast-, endometrial-, and ovarian cancer. Blocking the estrogen pathway has been a pivotal part in these hormone-dependent cancers. A few studies have reported the interaction between fucoidan and estrogen. In a study on the effect of fucoidan (*F. vesiculosus* extract, FVE) in vitro on different female cancer cell lines (breast-, ovarian-, uterine-, endometrial carcinoma); FVE proved to inhibit estrogen receptor (ER) activation by E2 and inhibition of E2 synthesis [91]. The overall effect was the inhibition of proliferation of breast and ovarian cancer cell lines, in both ER positive and ER negative cells. Fucoidan extract (*Cladisphon navae-caledonia*) was combined with three chemotherapeutic drugs in clinical use (tamoxifen, paclitaxel, and cisplatin) in breast cancer cells. A highly synergistic effect occurred with all three drugs tested. Importantly, the fucoidan extract protected healthy human fibroblast from the cytotoxic side effects [172].

There is sufficient data that support the interaction of fucoidan with estrogen and estrogen receptors. However, the direct inhibition of ER activity and the aromatase activity has yet not been reported and needs to be further explored.

## 3. Discussion

Many studies have reported on the different pathways involved in cancer that are affected by fucoidan. Interpretation is hampered by: structural diversity of fucoidan from different species (i), different extraction and purification techniques (ii), cell specificity of fucoidan (iii), and limitations of studies on fucoidan (iv). Therefore, it remains challenging to extrapolate final conclusions to other cancer types or other brown algae species.

### 3.1. Structure and Bioactivity

Fucoidan consists of a backbone of heterogeneous sulfated fucans, with complex branching of neutral sugars and sulfate groups. Due to the complexity, it is difficult to study it as a single molecule. Most studies limit themselves to highly purified fractions, hampering understanding the bioactivity of the whole molecule.

The structural diversity between species makes it challenging to establish a general or ‘basic’ structure for fucoidan molecules. Fucoidan, being a natural product, gives it some favorable characteristics already mentioned earlier this review. However, it also raises the trouble of having inconsistent products due to seasonal differences in growth conditions of the seaweed. Furthermore, the choice of harvest season is important for the final compound [41]. The monosaccharide composition has been connected with bioactivity on multiple occasions, but it is not clear if it contribute to their structure or that they just enable the binding of more sulfate groups [16].

The molecular weight plays a dual role concerning the bioactivity. In general, HWMF has shown more effectiveness than LMWF. However, LMWF has more favorable pharmacokinetically properties. A solution could be to develop LMWF, increasing the absorption rate [50]. However, there are concerns about the decreased bioactivity of LMWF, stressing the importance of the purification method. Purification by gamma-irradiation retains the bioactivity, by preventing the loss of branching. Improving the purification method of fucoidan would therefore increase the potential use of fucoidan-based treatment. Bioactivity could be further improved by modifications of the sulfate groups [53,55,56].

There is widespread consensus on the importance of the sulfate groups to the bioactivity. This has led to studies where fucoidan was modified to yield more sulfate groups. This could be a strategy to modify fucoidan molecules in order to increase the bioactivity and thus, the chance of inducing a clinical significant response.

The toxicity of fucoidan seems to be low for humans. Fucoidan is considered to be a food supplement and not a drug, so it is not regulated by the FDA, explaining the lack of clinical trials. However, the two ongoing clinical trials, could give more insight into the biodistribution of fucoidan in humans, and add data on toxicity and side-effects.

### 3.2. Cellular Mechanisms in Relation to Fucoidan Anti-Cancer Activity

We established now that fucoidan is able to interact with the PI3K/AKT signaling pathway, targeting it via multiple regulators. Due to the central role of the PI3K/AKT pathway, inhibition leads to both reduced cell proliferation and reduced migration of cancer cells.

We identified PTEN as a target for fucoidan-mediated inhibition of the PI3K/AKT signaling pathway, however this mechanism is not fully PTEN-dependent. The other involved proteins have yet to be discovered. The interaction with PTEN is relevant, as it is a common tumor suppressor. However, as PTEN is often mutated in cancer cells, it is questionable if fucoidan can stimulate it in the mutated form [192]. Moreover, the inhibitory effect on this pathway appears to be insufficient in cancer cells with a constitutively active GTP receptor, to have a significant, fucoidan-mediated, inhibitory effect on cell proliferation [114,115].

The MAPK pathway is associated with growth, proliferation and metastasis. There are several studies confirming the inhibitory effect of fucoidan on the protein levels of phosphorylated ERK1/2. Due to the fact that the reduction of ERK1/2 was often associated with reduction of other cellular pathways, it remains unclear if ERK1/2 is essential in fucoidan-mediated inhibition. The ERK1/2-mediated reduction of MMP-1 by fucoidan (*C. costata*) is interesting, proving the preventive potential of fucoidan in case of UVB irradiation [127]. This property of fucoidan has been reported before and could be used in the prevention of skin cancer [50].

The role of p38 MAPK is more modest, compared to the role of ERK1/2 and AKT in fucoidan-mediated effects. Both the reduction and stimulation of phosphorylated p38 MAPK leads to the induction of apoptosis [87,132,134].

The caspase pathway has been mentioned several times in fucoidan-mediated apoptosis. Pro-apoptotic members of the Bcl-2, regulated by the PI3K/AKT pathway, are able to induce the caspase pathway by releasing Cyt c from the mitochondria [193,194]. Tumor necrosis factor (TNF) and TNF-related apoptosis-inducing ligand receptor 1 (TRAIL) are transmembrane receptors (death receptors), able to induce the caspase pathway, upon activation [195]. It is not yet clear if fucoidan is able to activate the caspase pathway via these receptors, however some studies suggests this mechanism [145]. The interaction between the death receptors and fucoidan could be another significant mechanism of fucoidan-mediated apoptosis. Due to the regulation of Bcl-2 by the PI3K/AKT pathway, it is not known if fucoidan can interact directly with this side of the caspase pathway.

### 3.3. Receptors in Relation to Fucoidan Anti-Cancer Activity

The most relevant receptors for fucoidan activity are TGF-β, VEGF and ER. Since fucoidan is a large polysaccharide that has limited capability to cross the cell membrane, the impact of fucoidan on the cellular pathways is directly linked to fucoidans specificity for certain transmembrane receptors.

Transforming growth factor-beta (TGF-β) plays a dual role in tumorigenesis and is often upregulated in metastatic cancer cells. A potential fucoidan-mediated mechanism is the accelerated ubiquitin-dependent degradation of TGF-β receptors, by stimulating Smurf2 [150,154]. The bone morphogenetic proteins (BMPs) is a subfamily, belonging to the family of TGF-β, of cytokines involved in bone transformation. The intracellular pathway is similar to TGF-β [196]. The dual role of TGF-β and BMP in tumorigenesis has been reported by various studies [197]. TGF-β and BMP signaling were both reduced in cancer cells, after fucoidan treatment, while activated in healthy cells. These properties support fucoidan as an additional therapy to existing chemotherapy to improve treatment effect and reduce side effects. At the moment, there is limited evidence pointing out a synergistic effect of fucoidan, in reference to TGF-β and BMP. However, this deserves further exploration.

Fucoidan from various seaweeds is able to inhibit VEGF expression in different cancer cell lines. Importantly, the inhibition of fucoidan on VEGF signaling in multiple in vivo experiments is significant for fucoidans potential [168,169]. VEGF signaling is heavily embedded in the FAK and PI3K/AKT pathway, making the interaction with fucoidan a significant contribution in creating a clinical meaningful effect. It was further confirmed that the sulfate groups are important for fucoidans bioactivity, because a desulfated fraction failed at inhibiting tube formation [167], provoking more interest in modifying fucoidan molecules by adding more sulfate groups.

The supposed inhibitory effect of fucoidan on estrogen synthesis and ER activation, is interesting. Combatting cancer by blocking the estrogen pathway has been a pivotal method, however, has been criticized due to severe side effects [198]. The reported minimal effects of fucoidan on healthy cells and the ability to inhibit estrogen pathways, candidates fucoidan as a potential drug for estrogen-dependent cancer. Unfortunately, there are only few studies concerning this topic. A fucoidan extract from *F. vesiculosus* has been proven to inhibit the binding of estrogen to its receptor [91]. However, the exact mechanism is not fully understood. More elaborate studies are necessary to examine the potential of fucoidan (extract) in the treatment of estrogen associated cancer types.

The synergistic effects of fucoidan with other drugs, has been mentioned several times in the course of this review. When fucoidan was combined with various chemotherapy drugs, there was a synergistic effect noted on MDA-MB-237 cell lines, increasing the expression of pro-apoptotic biomarkers [199]. Various studies have also confirmed that fucoidan treatment does not interfere with the pharmacokinetics of other drugs [200]. Furthermore, fucoidan appears to increase the efficacy of some chemotherapeutics [46]. This could be an extra use of fucoidan in cancer treatment. However, due to the limited studies on this topic, it is challenging to assess the importance of this characteristic of fucoidan.

### 3.4. Clinical Significance

Fucoidan is a multipotent molecule that is, able to affect multiple pathways and types of receptors. Establishing a general mechanism of action is challenging, due to the limited knowledge and understanding of the different pathways and receptors. With the identification of PTEN as a target for fucoidan, we obtain more insight in the cellular mechanism of action of fucoidan. The discovery of PTEN as a target of fucoidan could stimulate more dedicated research into the precise mechanics of this interaction. Until now, there are no studies on the interaction of fucoidan with PTEN. That could be a possible next step to establish the importance of PTEN. However, the interaction with PTEN is just one of the many pathways and proteins that are affected. Despite the increased interest in fucoidan, we still lack the knowledge to grasp its effect on cellular mechanism. In some studies, fucoidan showed a dual role in its effect on the MAPK and PI3K/AKT pathway, resulting in either the stimulation or the inhibition of the pathways. In either case, it always led to apoptosis or at least a decrease in proliferation. This would suggest that the effects of fucoidan are more complex than we now know.

Targeting, of the growth-involved receptors by fucoidan could be useful in cancer treatment. Often, these receptors are overexpressed on the cancer cells. Mapping all the involved receptors is a valuable strategy to gain knowledge of all its interactions. However, the effect on the cellular mechanics remains unclear.

Studies have repeatedly shown that the structure of fucoidan is the most important determinant of its effect. The biggest challenge lies in the identification of the parts of fucoidan that are most active, so that the most optimal form of fucoidan can be employed in further treatment. Thus, understanding the structure could help us potentiate the compound to make it more suited for treatment. The structural variety also make it challenging to interpret data and extrapolate between multiple studies or cell lines.

### 3.5. Future Perspectives

There is clear evidence that fucoidan is able to inhibit the growth and proliferation of cancer cells. Therefore, fucoidan is a very promising candidate for future anticancer therapy. There seems to be no side-effects on healthy cells, it can interact with multiple targets increasing the change of producing a clinical meaningful effect, and it has wide-spread effect on many cancer types [201].

Despite the numerous studies on this topic, there are still very few clinical trials completed or planned. This is likely the result of a lack of comparative studies on, for instance, on specific fucoidan species. Instead, a lot of studies are examining different types of fucoidan on different cancer cell lines. This makes it difficult to establish a general mechanism of action of a specific type of fucoidan. Furthermore, there is relatively little known about the absorption, distribution, and excretion of fucoidan.

The best-studied fucoidan is the one derived from *F. vesiculosus*, that showed to negatively regulate the growth and proliferation of different types of cancer. The structure is also well understood. *F. vesiculosus* seems, for now, the ideal candidate to be further studied and eventually, tested in a clinical study. A good start will be the absorption and excretion properties of *F. vesiculosus* in humans. Furthermore, the development of LMWF, without the loss of bioactivity, of *F. vesiculosus* is interesting. This LMWF could then be used in clinical trials in an attempt to obtain more insight in the pharmacokinetics. The obtained data can then be used to eventually further transform the molecular characteristics of fucoidan, which has already been proven to be a potential way to improve bioavailability and bioactivity.

## 4. Methods

An electronic search was performed to identify relevant articles from the online databases Medline and Web of Science from inception until 15 October 2018. Search terms included ‘fucoidan’, ‘sulfated polysaccharides’, ‘cancer’ and ‘pharmacokinetics’. Citation lists were manually searched for other relevant articles. The complete search strategy is described in Appendix A.

## 5. Conclusions

Marine natural products are interesting as potential sources for cancer treatment. The amount of diverse bioactive compounds enlarges the chance of finding a suitable molecule with the right biochemical properties.

It is now well established that fucoidan interacts with multiple pathways, including PI3K/AKT and MAPK, and caspase pathway. Furthermore, various studies have confirmed the interaction of fucoidan with TGF-β, VEGF, BMP, and estrogen receptor. The omnipotent character of fucoidan increases the chance of producing clinically meaningful effects in future trials. However, the structural variety of the different types of fucoidan makes it challenging to draw conclusions about fucoidans bioactivity. The number of clinical trials in this topic is limited, resulting in restricted knowledge about fucoidans pharmacokinetics, safety, dosage, and potential interactions with other drugs. To prove fucoidans potential utility in cancer treatment, there is a need for standardization of purification method, focus research and development of LWMF of one promising seaweed species (*F. vesiculosus*), and the conduction of further clinical trials. In summary, based on the results obtained in preclinical studies fucoidan has anti-neoplastic properties, and considered to have a promising future in cancer treatment.

## Figures and Tables

**Figure 1 marinedrugs-17-00032-f001:**
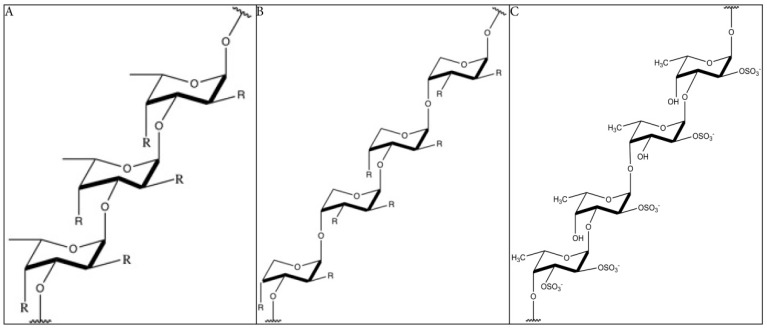
The backbone structure of fucoidan (simplified). (**A**): Structure of type 1 fucoidan molecules with a backbone of (1 → 3)-linked α-l-fucopyranose residues. The ‘R’ can be a monosaccharide or a sulfate group. (**B**): Structure of type 2 fucoidan molecules with a backbone alternating (1 → 3)-linked α-l-fucopyranose and (1 → 4)-linked α-l-fucopyranose residues. The ‘R’ can be a monosaccharide or a sulfate group (**C):** Structure of fucoidan from *F. vesiculosus*, with a backbone of alternating (1 → 3)-linked α-l-fucopyranose and (1 → 4)-linked α-l-fucopyranose residues and the presence of sulfate groups on both *O*-2 and *O*-3 [21,22,30]. R = uronic acid/rhamnose/glucose/galactose/xylose/mannose/arabinose/ribose/glucuronic acid (common found monosaccharides in fucoidan) [31,32,33,34,35,36,37,38,39,40].

**Figure 2 marinedrugs-17-00032-f002:**
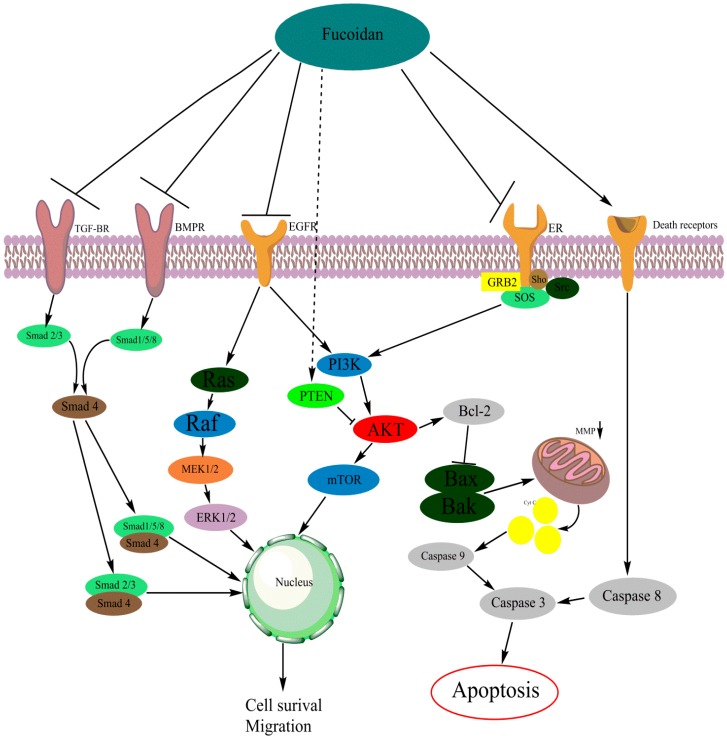
Mechanism of fucoidan-mediated inhibition on cellular pathways and receptors [121,136]. Figure 2 was made with the software Chemdraw (PerkinElmer Informatics, Cambridge, MA, USA).

**Table 1 marinedrugs-17-00032-t001:** The known inhibitory effects of different types of fucoidan on different types of cancer cells.

Cancer Type	Cell Line	Fucoidan	Mechanism	Research Methods	References
Breast cancer	MDA-MB-231 and MCF-7	*Fucus vesiculosus*	Inhibiting proliferation and metastasis	In vitro	[153]
	MCF-7	*Fucus vesiculosus*	Inhibiting proliferation, inducing cell cycle arrest, and inducing apoptosis	In vitro	[173,174]
	MDA-MB-231	*Fucus vesiculosus*	Inhibiting proliferation and inducing apoptosis	In vitro	[174]
	MDA-MB-231 and MCF-7	*Sargassum hemiphyllum*	Inhibiting proliferation, inducing cell cycle arrest, and inducing apoptosis	In vitro	[175]
	MDA-MB-231 and MCF-7	*Fucus vesiculosus* (extract)	Inhibiting proliferation and inducing apoptosis	In vitro	[91]
	MCF-7	Not stated (supposed *Fucus vesiculosus*)	Inhibiting proliferation and inducing apoptosis	In vitro	[176]
	MCF-7	*Fucus vesiculosus* (derivatives)	inhibiting proliferation	In vitro	[57]
	T47D	*Fucus vesiculosus* (extract)	Inhibiting proliferation and inducing apoptosis	In vitro	[91]
B-cell lymphoma	HS-sultan and IM-9	*Fucus vesiculosus*	Inhibiting proliferation and inducing apoptosis	In vitro	[122]
	DLBCL lines	*Fucus vesiculosus*	Inhibiting proliferation, inducing cell cycle arrest, and inducing apoptosis	In vitro and In vivo	[177]
	Raji cells	*Saccharina latissima and Fucus vesiculosus*	Inhibiting metastasis	In vitro	[178]
	BCBL-1 and TY-1	*Cladosiphon okamuranus*	Inhibiting proliferation and inducing apoptosis	In vitro and In vivo	[179]
T-cell lymphoma	MOLT-4	*Fucus vesiculosus*	Inhibiting proliferation and inducing apoptosis	In vitro	[122]
	MT-2, MT-4, HUT-102, and MT-1	*Cladosiphon okamuranus*	Inhibiting growth and inducing apoptosis	In vitro	[180]
Fibroblastic sarcoma	HT 1080	*Cladosiphon novae-caledoniae* (extract)	Inhibiting metastasis	In vitro	[181]
Uterine sarcoma	HeLa	*Cladosiphon novae-caledoniae* (extract)	Inhibiting metastasis	In vitro	[181]
	HeLa	*Fucus vesiculosus* (derivatives)	inhibiting proliferation	In vitro	[57]
	HeLa	*Fucus vesiculosus* (fractions)	Inhibiting proliferation, growth, and inducing apoptosis	In vitro	[182]
	MES-SA	*Fucus vesiculosus* (extract)	Inhibiting proliferation and inducing apoptosis	In vitro	[91]
Lung cancer	LLC1	*Fucus evanescens*	Inhibiting proliferation and metastasis	In vitro	[183]
	LLC1	*Sargassum sp. And Fucus vesiculosus*	Inhibiting proliferation and inducing apoptosis	In vitro	[184]
	A549	*Fucus vesiculosus*	Inhibiting metastasis	In vitro	[92]
	LLC1, A549, and CL1-5	*Fucus vesiculosus*	Inhibiting proliferation, metastasis and inducing apoptosis	In vitro and In vivo	[154]
	LLC1	*Fucus vesiculosus*	Inhibiting proliferation and metastasis	In vitro and In vivo	[70]
	A549	*Sargassum fusiforme* (sulfated extract)	Inhibiting proliferation and metastasis	In vitro and In vivo	[168]
	A549, LLC1, and CL1-5	*Fucus vesiculosus and Saccharina Japonica*	Inhibiting proliferation and inducing apoptosis	In vitro and In vivo	[124]
Hepatocellular carcinoma	HuH-6	*Cladosiphon okamuranus*	Inhibiting biotinidase activity	In vitro	[185]
	Huh-6, HUH-7, SK-Hep1, and HepG2	*Sargassum hemiphyllum*	Inhibiting proliferation and metastasis	In vitro	[152]
	HepG2	*Fucus vesiculosus* (fractions)	Inhibiting proliferation, growth and inducing apoptosis	In vitro	[182]
	Hca-F	*Kjellmaniella crassifolia*	inhibiting proliferation	In vitro	[186]
Colorectal cancer	HCT-15	*Fucus vesiculosus*	Inhibiting proliferation and inducing apoptosis	In vitro	[87]
	HT-29 and HCT-116	*Fucus vesiculosus*	Inhibiting proliferation and metstasis	In vitro	[145]
	HCT-116	*Fucus vesiculosus*	Inhibiting proliferation and inducing apoptosis	In vitro	[174]
	HT-29	*Fucus vesiculosus*	Inhibiting proliferation, inducing cell cycle arrest, and inducing apoptosis	In vitro and In vivo	[96]
	HCT-116, HT-29, and WiDr	*Fucus evanescens*	Inhibiting colony formation and growth	In vitro and In vivo	[187]
	HCT116	*Sargassum hemiphyllum* (LMWF, oligo-fucoidan)	Inhibiting proliferation, inducing cell cycle arrest, and inducing apoptosis	In vitro and In vivo	[188]
	HCT116	*Fucus vesiculosus*	Inhibiting proliferation, inducing cell cycle arrest, and inducing apoptosis	In vitro	[189]
Keratinocytes	HaCaT	*Costaria costata*	Inhibiting metastasis	In vitro	[127]
Melanoma	B16	*Sargassum sp. And Fucus vesiculosus*	Inhibiting proliferation and inducing apoptosis	In vitro	[184]
Bladder cancer	5637 and T-24	Not stated	Inhibiting proliferation, growth and inducing cell cycle arrest	In vitro	[94]
	T-24	*Sargassum hemiphyllum*	Inhibiting angiogenesis	In vitro and In vivo	[45]
	5637	Fucus vesiculosus	induction apoptosis	In vitro	[88]
Plasma cell myeloma	RPMI8226 and U266	Not stated	Inhibiting angiogenesis	In vitro	[190]
Leukemia	U937, HL60, K562, THP1	*Fucus vesiculosus*	Inhibiting proliferation and inducing apoptosis	In vitro	[132]
	NB4, HL60, and K562	*Fucus vesiculosus*	Inhibiting proliferation and inducing cell cycle arrest	In vitro and In vivo	[69]
	SKM-1	Not stated (supposed *Fucus vesiculosus*)	Inhibiting proliferation and inducing apoptosis	In vitro	[90]
Stomach cancer	MKN45	*Cladosiphon okamuranus*	Inhibiting proliferation and inducing cell cycle arrest	In vitro	[133]
Pancreatic cancer	MiaPaCa-2 and Panc-1	*Turbinaria conoides*	Inhibiting proliferation, metastasis and inducing apoptosis	In vitro and ex vivo	[49]
Ovarian cancer	OVCAR-3	*Fucus vesiculosus* (extract)	Inhibiting proliferation and inducing apoptosis	In vitro	[91]
Endometrium carcinoma	HEC-1B, RL95-2, and AN3CA	*Fucus vesiculosus* (extract)	Inhibiting proliferation and inducing apoptosis	In vitro	[91]
Prostate cancer	DU-145	Not stated (supposed *Fucus vesiculosus*)	Inhibiting proliferation and metastasis	In vitro and In vivo	[191]
Osteosarcoma	MG63	*Fucus vesiculosus*	Inhibiting angiogenesis	In vitro	[169]

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
