# Peer review of "Fucoidan Structure and Activity in Relation to Anti-Cancer Mechanisms"

_marinedrugs, 2019, doi:10.3390/md17010032_

Reviewer 1 Report

The potential anti-cancerogenic actions of fucoidans is an interesting and important topic. Despite some attention in the literature (also in Marine drugs) the subject area is worthy of deep investigation and hence this review is justified.  However, major revision is required as follows before the manuscript can be endorsed for publication. 1.  The title is misleading as an in depth "investigation" to answer this should include a lot of other things such as required clinical trials, approval procedures etc., rather, the paper is on "fucoidan chemistry and putative anti-cancer mechanisms". The title should be changed accordingly. 2.  Figures 1-2-3 can be assembled into one Figure!  However, please note that detailed fucoidan structures have been published many times already, and the backbone(s) shown do not cover "fucoidans" especially not those from Sargassum that are more complex in both their backbone and sidechains (see recent papers in Marine drugs), and wrt sulfatation  than the Fucales fucoidans; It is fair enough to only draw the classic alfa1,3-1,4-linked L-fucosyl-linked backbones - There is a whole section about sulfatation, but still, the sulfatation should be detailed and discussed more in relation to the anti-cancerogenic effects as the sulfatation has previously been suggested to play a positive role. 3. The review would be more focused if the authors would refrain from considering "bioactivity"...please focus solely on anti-cancer mechanisms, and consider a) preventive action vs. cancer treatment action mechanisms, and please distinguish more carefully between data from in vitro/ex vivo cell experiments and in vivo data, I suggest that TABLE 1 is split into two Tables: One with in vitro cell assay data, one with in vivo data. 4. Focus more on illustrating the mechanisms in more detail (Figure 4 is too primitively drawn). Consider including more reaction schemes for the mechanisms - e.g. section 2.3.1. should also include visual schematics of pathways- 5. Other papers have suggested pathways in Marine Drugs, loks like you overlooked the apoptosis caspase-3 suggested pathway (Mar. Drugs 2011, 9, 2605-2621)? 6. A dosage-response discussion is lacking. In the conclusions it just says "The number of clinical trials in this topic is limited, resulting in limited knowledge about fucoidans pharmacokinetics, safety, dosage, and potential interactions with other drugs" - This might be true, but there are dosage-response data available in the literature for animal studies and these should be more carefully considered in the review.   7. Appendix 1 is obsolete.

Author Response

We thank the reviewer for the suggestion on our manuscript. See the attached word document for our response.

Reviewer 2 Report

Corrections needed

Introduction
line 40 - Add the name of the class of brown macroalgae: brown algae (Phaeophyceae)
line 42 - Comment: not all algae are marine, some of them are present in freshwater
line 43 - Comment: The dominance of brown algae on the coast is relative, since in fact the greatest number of species belongs to the Rhodophyta phylum, the red algae. The dominance of the Phaeophyceae refers to their  biomass and not in relation to the number of species.
line 61 - This manuscript is a review and not a paper on experimental work, so it should not have a heading called "Results"
line 62 - 2. Structure of Fucoidan
line 66 - According to the rules of taxonomy, the only categories written in italics are the "genus" and the "species", so the word "Fucales" should be written in normal letters.
line 88 - The valid name is Himanthalia elongata, so write "... fucoidans from Himanthalia elongata (formerly H. lorea)..."
line 114 - 2.1
and so on
line 131 - "...of the sulfate groups in Saccharina cichorioides (formerly Laminaria cichorioides), F. evanescens and Saccharina japonica (formerly Laminaria japonica) are in axial positions..."
line 147 - "LMWF (Sargassum hemiphyllum ) was able to..."
line 181 - (Saccharina japonica - formerly L. japonica)
line 259 - (Saccharina japonica - formerly L. japonica)
line 335 - (Saccharina japonica - formerly L. japonica)
line 353 - the word "extract" not in italics
line 402 -Table 1 - Please the species names in italics (all) and Saccharina japonica (formerly L. japonica) ...  Kjellmaniella crassifolia (formerly Saccharina sculpera)

Author Response

(The authors gave the same response as above.)

Reviewer 3 Report

Thank you for reviewing the properties of fucoidan in oncology.  As you have described, multiple pathways are affected, and in some cases, fucoidan appears to assist in the effectiveness of common chemotherapy agents.  Another aspect that should perhaps be commented on is the effect of fucoidan on ordinary primary cells (Non cancer cells), as this is also important.

You comment about the need for absorption and excretion studies of Fucus derived fucoidan in humans. Because fucoidan is considered to be a 'food supplement' and not a 'drug',  these kinds of studies have not been done.  It  may be worthwhile to make a comment on the regulatory status of 'fucoidan' within the text.

There is a  problem with text  at line 552/553. 

Author Response

(The authors gave the same response as above.)

Reviewer 4 Report

This review article summarized the anti-cancer effects of fucoidan at the several aspects. There are some points to be corrected for the publication.

The contents of Discussion are overlapping with Result. I am not sure they should be separated.

There are many misspelling to be corrected or checked.

Misspelling

Line 120 – content as (and)

Line 122 – are (is)

Line 123 – 12,4 kDa (12.4 kDa) All molecular weights were marked like this.

Line 226 – let to (led to)

Line 242 – one of the different interactions fucoidan has on the PI3K/AKT pathway. (This sentence should grammatically be checked, and the meaning is not clear.)

Line 436 – immune, the meaning is obscure and not appropriate, ‘insensitive or resist’ might be better

Line 473 – his (its)

Line 477 – inhibited (inhibit)

Line 496 – are is (is)

Line 499 – who (that)

Line 500 – his (its)

Line 507 – his (its)

Author Response

We thank the reviewer for the suggestion on our manuscript. See the attached word document for our response.

Round  2

Reviewer 1 Report

The authors have conscientiously revised the manuscript in response to the comments and the revision has improved the manuscript. However, a few important mistakes still need to be corrected before the manuscript can be  accepted.  

1. Line 44-46: The following statement is wrong/wrongly cited "..In reference to biomass, brown algae are the dominant organism in coastal regions...". It is to general, The wording should be "in many coastal regions..."  (ref [7]).

2. Figure 1: It has helped to condense the structural drawings into Fig 1 A, B, C, but Fig 1C is misleading with too many sulfate groups at position C3. The structure of the F. vesiculosus fucoidan is rather that the L-fucosyl residues are sulfated at position C2 and/or C4, but rarely at C3.  

3. Figure 1: References are missing(?)

4. Figure 1, and several places in the text: please refrain from writing "a" for the Greek "α" (alfa) when it comes to referring to the bonds, it must be "α-L-fucopyranose residues".

Minor

5. For the authors' affiliations: Affiliation 3 and 5 seem to be the same, so no reason to have two different indexes?

6. The homespun abbreviation "SAR" for "structure-activity relationship" is not used beyond the 2 places where it is explained in the text (line 58-59, line 66), hence, not needed.  

7. Legend Figure 1 "The ‘R’ can be a sugar"; please change the wording "sugar" to the specific monosaccharides possible.

8. All organism names of the macroalgae must be in italics. This is missing in many, many of the reference titles.

9. Still think that Appendix 1 is obsolete, but it can stay if the Editor think so - Although it is "relevant" to see the search profile, it is really the interpretation of the references which matters, and with peer review the "search" span should be secured. This is why the search profile is not needed.  

Author Response

We thank the reviewer for the comments. See attached our response
